# The Role of Human-Induced Pluripotent Stem Cells in Studying Cardiac Channelopathies

**DOI:** 10.3390/ijms252212034

**Published:** 2024-11-08

**Authors:** Merima Begovic, Luca Schneider, Xiaobo Zhou, Nazha Hamdani, Ibrahim Akin, Ibrahim El-Battrawy

**Affiliations:** 1Institute of Physiology, Department of Cellular and Translational Physiology, Ruhr-University Bochum, 44801 Bochum, Germany; merima.begovic@ruhr-uni-bochum.de (M.B.); luca.schneider@ruhr-uni-bochum.de (L.S.); nazha.hamdani@ruhr-uni-bochum.de (N.H.); 2Institut für Forschung und Lehre (IFL), Molecular and Experimental Cardiology, St. Josef Hospital, Ruhr-University Bochum, 44791 Bochum, Germany; 3Cardiology, Angiology, Haemostaseology, and Medical Intensive Care, Medical Centre Mannheim, Medical Faculty Mannheim, Heidelberg University, 68167 Mannheim, Germany; xiaobo.zhou@medma.uni-heidelberg.de; 4Key Laboratory of Medical Electrophysiology of Ministry of Education and Medical Electrophysiological Key Laboratory of Sichuan Province, Institute of Cardiovascular Research, Southwest Medical University, Luzhou 646000, China; 5Department of Physiology, Cardiovascular Research Institute, University Maastricht, 6229HX Maastricht, The Netherlands; 6HCEMM-SU Cardiovascular Comorbidities Research Group, Center for Pharmacology and Drug Research & Development, Department of Pharmacology and Pharmacotherapy, Intézet címe Semmelweis University, 1089 Budapest, Hungary; 7Department of Cardiology and Rhythmology, St. Josef Hospital, Ruhr University, 44791 Bochum, Germany

**Keywords:** sudden cardiac death, Brugada syndrome, catecholaminergic polymorphic ventricular tachycardia

## Abstract

Cardiac channelopathies are inherited diseases that increase the risk of sudden cardiac death. While different genes have been associated with inherited channelopathies, there are still subtypes, e.g., catecholaminergic polymorphic ventricular tachycardia and Brugada syndrome, where the genetic cause remains unknown. Various models, including animal models, heterologous expression systems, and the human-induced pluripotent stem-cell-derived cardiomyocytes (hiPSCs-CMs) model, have been used to study the pathophysiological mechanisms of channelopathies. Recently, researchers have focused on using hiPSCs-CMs to understand the genotype–phenotype correlation and screen drugs. By combining innovative techniques such as Clustered Regularly Interspaced Short Palindromic Repeats/Clustered Regularly Interspaced Short Palindromic Repeats associated protein 9 (CRISPR/Cas9)-mediated genome editing, and three-dimensional (3D) engineered heart tissues, we can gain new insights into the pathophysiological mechanisms of channelopathies. This approach holds promise for improving personalized drug treatment. This review highlights the role of hiPSCs-CMs in understanding the pathomechanism of Brugada syndrome and catecholaminergic polymorphic ventricular tachycardia and how these models can be utilized for drug screening.

## 1. Cardiac Channelopathies

Channelopathies are diseases that result from dysfunctional ion channels due to either genetic or acquired pathological factors. Genetic channelopathies occur as a result of variations in the genes encoding the regulatory or pore-forming subunits. On the other hand, toxins, drug exposure, and acquired disorders can all contribute to the development of acquired cardiac channelopathies. The most common cardiac channelopathies include long QT syndrome (LQTS) and Brugada syndrome (BrS), in addition to conditions like early repolarization syndrome, catecholaminergic polymorphic ventricular tachycardia (CPVT), short QT syndrome (SQTS), and isolated progressive cardiac conduction disease. In this review, we focus on BrS and CPVT due to their rarity and their significant insight into the genetic and molecular mechanisms underlying less understood channelopathies.

LQTS and SQTS syndrome are significant genetic cardiac channelopathies that markedly increase the risk of arrhythmias and sudden cardiac death (SCD). LQTS is characterized by delayed ventricular repolarization, resulting in a prolonged QT interval, which predisposes affected individuals to life-threatening arrhythmias such as torsades de pointes. In contrast, SQTS is distinguished by accelerated repolarization, leading to a shortened QT interval and heightened susceptibility to both atrial and ventricular arrhythmias. Both conditions arise from gene mutations that encode ion channels, ultimately disrupting cardiac electrophysiology and rhythm stability.

Recent studies using hiPSC-derived cardiomyocytes have elucidated the mechanisms that are underlying LQTS and SQTS. These stem cell models provide a valuable platform for investigating patient-specific gene mutations and the effects of pharmacological interventions, thereby enhancing our understanding of and treatment of these syndromes [1,2,3].

In the cardiovascular system, ion channels play a crucial role in various aspects of cardiac function, including rhythmicity and contractility. When there is an alteration in these ion channels, it increases the risk of atrial and ventricular arrhythmic events, which can predispose individuals to sudden cardiac death (SCD). Inherited arrhythmia syndromes, which encompass inherited cardiac channelopathies, account for more than 30% of SCD cases in young individuals without any underlying structural heart disease at a young age [4]. Distinct types of cardiomyopathies, encompassing both genetic and acquired variations, significantly contribute to the risk of SCD. Moreover, we recognize the critical role of iPSC-derived cardiomyocytes in enhancing our understanding of the mechanisms underlying SCD. The review article “Human Induced Pluripotent Stem-Cell-Derived Cardiomyocytes as Models for Genetic Cardiomyopathies” is cited to provide a foundational perspective on this topic, highlighting how these models can elucidate the underlying pathophysiology of SCD within the context of cardiomyopathies [5,6,7,8,9,10].

## 2. Long QT Syndrome and Short QT Syndrome

LQTS and SQTS are congenital cardiac channelopathies that disrupt the heart’s electrical repolarization phase, resulting in abnormal cardiac rhythms and an increased risk of SCD. In LQTS, delayed ventricular repolarization prolongs the QT interval, thereby elevating the risk of torsades de pointes, a potentially fatal ventricular arrhythmia. This condition is frequently attributed to mutations in genes such as *KCNQ1*, *KCNH2*, and *SCN5A*, which encode potassium and sodium channels.

In contrast, SQTS is characterized by an abnormally shortened QT interval due to accelerated ventricular repolarization, which is caused by gain-of-function mutations in potassium channels (KCNH2, KCNQ1, and KCNJ2), loss-of-function mutations in calcium channels (CACNA1C/CACNB2), or affecting chloride bicarbonate transporters encoded by *SLC4A3*. Such mutations predispose patients to atrial fibrillation, ventricular arrhythmias, and SCD. While LQTS results in delayed repolarization, SQTS facilitates its acceleration; both conditions can lead to significant arrhythmogenic disturbances.

A comprehensive understanding of these syndromes highlights the critical role of ion channels in maintaining cardiac rhythm. Furthermore, it emphasizes the challenges associated with diagnosing and managing these life-threatening conditions, which often necessitate individualized therapeutic strategies such as β-blockers, implantable cardioverter-defibrillator (ICD) implantation, or antiarrhythmic medications. Therefore, LQTS and SQTS are integral to advancing our understanding of cardiac electrophysiology and refining strategies for the prevention of SCD.

## 3. Brugada Syndrome

### 3.1. Clinical Background

Brugada syndrome (BrS) is one of the most common inherited arrhythmogenic syndromes. It was first diagnosed in 1992 and is characterized by a typical electrocardiogram (ECG). This pattern includes a right bundle branch block and persistent elevation of the region between the end of ventricular depolarization and beginning of ventricular repolarization (ST segment) in the right precordial that leads V_1_ to V_2_ and SCD [11].

BrS affects 0.5 people per 1000 people worldwide, with Southeast Asia having the highest prevalence at 3.7 per 1000 [12]. The majority of patients are male, suggesting a possible role of hormones in the pathophysiology of BrS and the distribution of ion channels in the heart [13]. A study found that the proportion of females among younger patients was significantly higher than the overall proportion. The majority of patients were identified through family screening and have the highest risk of SCD [14]. Newly identified patients are usually asymptomatic or have a history syncope, with only a small proportion experiencing cardiac arrest [15].

### 3.2. Genetic Background

Ventricular fibrillation is more common in younger patients and those with pathogenic/likely pathogenic variants in the *SCN5A* gene [16]. The inducibility of ventricular tachyarrhythmia has no impact on future cardiac events in asymptomatic BrS patients [17].

The genetic background of BrS patients has been, and is still being, investigated. Numerous variants in the *SCN5A* gene coding the α-subunit of the cardiac sodium channel Na_v_1.5 have been identified [18]. It was discovered that *SCN5A* (600163) gene variants occurred in up to 30% of BrS cases. Other gene variants have also been related to the phenotype of BrS. Genes that have been found are *SCN10A* (604427), *SCN1B-3B* (600235, 601327, 608214), *GPD1L* (611778), *RANGRF* (607954), *SLMAP* (602701), *ABCC9* (601439), *KCNH2* (152427), *KCNE3* (604433), *KCNJ8* (600935), *KCNE5* (300328), *KCND3* (605411), *HCN4* (605206), *CACNA1C* (114205), *CACNB2* (600003), *CACNA2D1* (114204), *TRPM4* (606936), and *PKP2* (602861) [19]. A part of the variants cause less or abnormal Na_v_1.5, causing a decrease in peak cardiac sodium channel current. However, in a bevy of cases of patients carrying variants in the *SCN5A* or other genes, the causal relationship has not been proved yet [20,21].

Genes associated with the BrS can be distinguished into four main groups as described in Figure 1. Sodium channel genes or genes that are molecularly linked to the sodium channel contain *SCN5A*, *GPD1L*, *SCN1B*, *SCN3B*, *RANGRF/MOG1*, *SCN10A*, *SLMAP*, *SCN2B*, and *PKP2* [22,23,24,25,26,27,28,29,30]. Plakophilin-2 (PKP2) plays a crucial role in cell–cell adhesion and is frequently mutated in inherited cardiac diseases, including arrhythmogenic right ventricular cardiomyopathy (ARVC/ACM) and BrS [22,31]. Notably, a phenotypic overlap exists between ARVC/ACM and BrS, suggesting that these conditions may represent different manifestations of a common underlying pathological process [32]. Cerrone et al. first reported cases of BrS without overt structural cardiomyopathy in patients carrying *PKP2* variants, indicating a potential link between these conditions [22]. While recent evidence-based studies have identified *SCN5A* as the primary gene associated with BrS, the role of PKP2 should not be overlooked [33,34]. Several preclinical studies in ARVC showed a significant decrease in the peak sodium channel current consistent with the finding in the BrS. The pleiotropic effects of PKP2 mutations provide crucial insights into how genetic alterations in different proteins interact at the connexome level, influencing both action potential dynamics and the structural integrity of the myocardium.

Genes involved in regulating calcium channels and linked to BrS are *CACNA1C*, *CACNB2*, *CACN2D1*, and *TRPM4* [35,36]. The genes *KCNH2*, *KCNE3*, *KCND3*, *KCNE5*, *KCNJ8*, and *SEMA3A* (603961) regulate potassium channel function [37,38,39,40,41,42]. Recent years have demonstrated that gene alterations unrelated to channelopathies can also contribute to the condition. Alterations in *ABCC9* and *HCN4* were found to be associated with BrS and could function on the molecular level as genetic triggers [43,44]. Further investigation is necessary to clarify the precise molecular functioning, which is still unclear.

The clinical management of BrS patients is limited. The only proven effective strategy for preventing SCD in symptomatic BrS patients is the use of an implantable cardioverter-defibrillator (ICD). In asymptomatic patients, a treatment algorithm is required in the referenced center to prevent over-ICD-implantation and to decrease the risk of ICD-related complications [45]. Our previous findings have shown that ICD therapy in BrS patients is linked to significant complications [46]. To address acute recurrent ventricular tachyarrhythmias, pharmacological therapy involving intravenously administrated isoproterenol medication and/or quinidine may be used [47]. However, drug treatment is not feasible and/or not effective in each asymptomatic patient. In addition, the in-compliance rate could be high. General lifestyle recommendations contain the avoidance of drugs that may induce ST-segment elevation in suitable precordial leads, i.e., the abstinence from cannabis, cocaine, and heavy alcohol consumption. When dealing with fever, antipyretics are the appropriate medication to use [48].

### 3.3. Pathomechanism

The exact pathophysiological causes of BrS are still unclear, and two theories are mainly under discussion. The repolarization theory is widely supported as the underlying mechanism for BrS. It suggests a disorder in repolarization, which is characterized by abnormal shortening of the epicardial action potential duration. This creates transmural voltage gradient, resulting in ST-segment increase, either by depressing or losing the action potential dome in the right ventricular epicardium and thereby probably leading to ST-segment increase [49,50]. The condition for the hypothesis lies in unaltered action potential shape by late sodium current (I_Na_) reduction [51]. Changes in the ECG right precordial leads are induced by a larger transient outward potassium current (I_to_) in the right ventricle [50]. The higher I_to_ density in men explains the higher disease prevalence in men compared to women [52].

The alternative theory is based on a depolarization disorder, with a focus on slower conduction through delayed depolarization [53]. The right ventricular outflow tract (RVOT) is delayed due to the right ventricular action potential. During the hatch phase in the cardiac cycle, the membrane potential becomes more positive in the right ventricle and functions as a source. This generates an intercellular current from the RVOT and a back current, resulting in ST-segment elevation in the right precordial leads [54]. Reciprocal events are recorded in the left precordial leads [55]. As a result, the cycle is now reversed and the membrane potential in the RVOT becomes more positive than in the right ventricle, functioning as the source [56].

## 4. Catecholaminergic Polymorphic Ventricular Tachycardia

### 4.1. Clinical Background

Catecholaminergic polymorphic ventricular tachycardia (CPVT) is a rare inherited arrhythmogenic syndrome which was first reported in 1975 [57]. In 1999, CPVT was identified as an inherited, autosomal dominant mode, with the first locus on chromosome 1q42-43 [58]. The same common genetic cause of CPVT is a variant in the Ryanodine receptor 2 (RyR2) gene, which is seen in the majority of cases [59]. CPVT is induced by emotional stress or exercise in patients without structural heart disease and may lead to adrenergic-induced bidirectional or polymorphic ventricular tachycardia [60]. The prevalence of the disease is up to 1 per 10,000 in Europe and may end in SCD. CPVT is found in children and adolescents and has poor prognosis for early syncopes [61]. The mortality rate is extremely high, up to 31% by the age of 30 if left untreated [62]. Additionally, patients at a high risk for SCD may require the insertion of an ICD [63]. In addition to medical and surgical interventions, lifestyle modifications play a crucial role in managing CPVT. Individuals diagnosed with CPVT are generally advised to avoid triggers such as strenuous physical activity and intense emotional stress, which can precipitate arrhythmic events [61].

### 4.2. Pathomechanism

At the molecular level, CPVT is characterized by genetic variants in two genes: the cardiac ryanodine receptor (cRyR) and cardiac calsequestrin 2 (CASQ2) [64]. RyR2 is a tetrameric protein located on the membrane of the sarcoplasmic reticulum (SR), which supplies ions during systole [65]. It is anchored to CASQ2 via satellite proteins [66]. Various other proteins, including calmodulin (CaM), FKBP12.6 (calstabin2), protein kinase A (PKA), phosphatase 1 (PP1), and phosphatase 2 (PP2), are connected to the cytoplasmic region or the luminal side [67]. The pathophysiology of CPVT involves disruptions in calcium homeostasis, which are caused by variations in key calcium-handling proteins, such cRyR2 and CASQ2. These variations interfere with normal calcium signaling in heart cell regulation, leading to abnormal Ca^2+^ release during sympathetic stimulation [68]. Specifically, variations in CPVT1 and CVPT2 result in calcium leakage from the SR, inducing delayed afterdepolarizations (DADs) through cytosolic calcium overload. The abnormally released Ca^2+^ from the SR activates the Na^+^/Ca^2+^ exchanger (NCX). This activation generates a transient inward current capable of producing DADs. If DADs reach the threshold potential, they can cause premature ventricular contractions and increase the vulnerability for arrhythmias. Additionally, early afterdepolarizations (EADs) can occur during the action potential plateau or repolarization phase, especially in case of prolonged action potential, which often overlaps with CPVT. In addition to variations in the RyR2 gene, other, less common genetic modifications associated with CPVT have been identified, involving genes such as *TRDN* (603283), the calmodulin gene family *CALM1* (114180), *CALM2* (114182), and *CALM3* (114183), *ANK2* (106410), and *TECRL* (617242) [69]. However, the significance of several other genes including *PKP2*, and *SCN5A* in the pathology of CPVT has been largely disproven. Furthermore, the high incidence of the *ANK2* gene variant in the general population makes it an unlikely primary contributor to CPVT etiology [69]. In 2016, a study provided the first insights into the role of *TECRL* variation in CPVT, showing a phenotype of CPVT [70]. More recently, a study using a knockout mouse model of the *TECRL* gene also revealed alterations in mitochondrial function [71]. Several pathogenic hypotheses propose the pathomechanism of CPVT. One theory suggests that there is a weakened binding affinity between mutant RyR2 and binding protein FKB12.6, causing the dissociation of FKBP12.6 after phosphorylation of RyR2 by PKA. The dissociation of FKBP12.6 leads to open channels, and Ca^2+^ can leak during diastole [72]. The store-overload-induced Ca^2+^ release (SOICR) theory assumes an SOICR threshold in the RyR2 mutant below the free Ca^2+^ in the SR, leading to a run over of Ca^2+^ from the SR [73]. Another theory suggests a defective intramolecular domain interaction. The variant interferes with the tight zipping of the molecular structure, leading to unzipping of the interdomain structure and resulting in leaking of Ca^2+^ from the SR [74]. A further possibility in addition to early afterdepolarization (EAD)/DAD is that CPVT patients might also experience ventricular fibrillation due to an irregular automaticity in the Purkinje fibers responsible for propagating cardiac impulses. This suggests that aberrant signal transmission within this system could promote a substrate for heart rhythm disorder [75]. An increase in catecholamine levels can trigger ventricular arrhythmias, which is a significant concern in CPVT patients due to their heightened response to sympathetic stimulation [76]. Some studies showed that variations affecting ß-adrenergic signaling pathway may be involved in CPVT [77]. Furthermore, overactivity of the calmodulin-dependent kinase II (CaMKII) pathway may contribute to CPVT in individuals carrying an RyR2 variant [78]. The chronology of the genes implicated in CPVT is shown in Figure 2. There are two distinct groups of genes linked to CPVT. All of the genes for the proteins that control the release of SR calcium during excitation-coupling are found in the leading group, including *RYR2* (180902), *CASQ2* (114251), *TRDN*, *CALM1*, *CALM2*, and *CALM3* [79,80,81,82,83,84]. Additional genes that encode proteins that cause CPVT but whose mode of action is unknown include *KCNJ2* (600681), *ANK2*, and *TECRL* [70,81,85].

## 5. Human-Induced Pluripotent Stem Cells (hiPSCs)

Human-induced pluripotent stem cells (hiPSCs) have opened new avenues for studying molecular mechanisms associated with cardiac ion channel abnormalities and have proved new opportunities for cardiovascular research. In the past, functional studies on specific variants have primarily relied on animal models. However, these studies have faced challenges due to differences in the time component, the gene expression profile, and physiology differences between species, which limit the validity of the data. The discovery of reprogramming somatic cells to generate hiPSCs was a significant milestone in research. Takahashi et al. were the first to generate hiPSCs by transforming human dermal fibroblasts using retroviruses carrying defined factors [86]. This breakthrough allowed researchers to easily obtain somatic cells from hair, blood, skin, fat, or oral mucosa, creating new opportunities for generating patient- and disease-specific cells. These somatic cells can be reprogrammed to a pluripotent state and differentiated into cardiomyocytes. HiPSCs can differentiate into myocytes with a cardiac-specific profile, making them ideal models for studying cardiac diseases, including cardiac channelopathies [87].

To generate hiPSC lines, somatic cells from donor samples such as skin, biopsy, blood, or urine samples are reprogrammed [86,88,89]. This process has been achieved using lentiviral particles carrying the reverse tetracycline-controlled transactivator (rtTA) and an inducible polycistronic cassette with the reprogramming factors octamer-binding transcription factor 4 (OCT4), sex-determining region Y-box 2 (also known as SOX2), Krüppel-like factor (KLF4), and cellular myelocytomatosis oncogene (c-MYC) [5,90,91,92,93]. The Yamanaka transcription factors are required to induce pluripotency, and lenti- or retroviral vectors have effectively delivered these factors. However, a drawback of these vectors is that their viral genome becomes incorporated into the host genome [94].

Episomal plasmids present a nonintegrating alternative to viral methods, thereby minimizing biosafety concerns. These plasmids facilitate the expression of reprogramming factors (OCT4, SOX2, NANOG, etc.) without the risk of genomic integration. Their episomal nature permits removal from the cells, thereby enhancing safety for clinical applications [95,96]. In addition, nonintegrating vectors such as Sendai virus vectors can be used [97,98,99]. Another option is the integration-free CytoTune-iPS 2.0 Sendai Reprogramming Kit, which includes the reprogramming factors OCT4, SOX2, KLF4, and c-MYC and does not result in viral genome integration [1,90,100,101].

Furthermore, mRNA transfection provides a vector-free approach that eliminates risks associated with residual vector traces. This technique also offers improved reprogramming speed and genomic integrity, rendering it suitable for the generation of clinical-grade hiPSCs. While blood-derived iPSCs show promise, their clinical application necessitates overcoming challenges related to mass production and reprogramming efficiency. By employing these nonintegrating techniques, it is possible to address biosafety concerns and enhance the clinical potential of hiPSC-derived cells [102].

## 6. Human-Induced Pluripotent Stem Cells Derived Cardiomyocytes (hiPSCs-CMs)

HiPSCs technology has sparked a surge in disease modeling, demonstrating its potential as a cellular model. The differentiation and maturation of hiPSCs to cardiomyocytes have significantly improved in recent years. However, the generation of quantitative, enriched hiPSC-cardiomyocytes (hiPSCs-CMs) remains a challenge due to the labor-intensive, time-consuming, and costly differentiation process. Restoring the contractile function of hiPSC-CMs has been a significant task, as their myofibril alignment is similar to that of fetal CMs [103]. Various protocols have been developed and examined to enhance the maturation of hiPSC-CMs towards the adult phenotype, with a focus on their electrophysiological properties. Consequently, these sophisticated protocols for hiPSC-CMs serve as excellent models for studying inherited channelopathies, such as BrS and CPVT, and investigating the electrophysiological alterations caused by ion channel anomalies [104]. Continual improvement and adjustment of the cardiomyocyte differentiation protocol have been carried out, from the three-dimensional differentiation of embryoid bodies with bone morphogenetic protein 4 (BMP4) and activin A to a monolayer-based approach utilizing the Wingless-related integration site/catenin beta-1 (Wnt/β-catenin)-signaling pathway. The monolayer-based approach is preferred due to its optimal conditions for the diffusion of the differentiating factors and highest differentiation efficiency, resulting in the production of many ventricular cardiomyocytes [105,106]. The differentiation process begins with the inhibition of glycogen synthase kinase 3β (GSK-3β), followed by the addition of a Wnt/β-catenin pathway inhibitor after 2–3 days of cultivation. After 7 to 8 days, the cells start beating spontaneously [64]. The differentiation of hiPSCs-CMs leads to a cell culture with a mixed type of cardiomyocytes, including ventricular-like, atrial-like, and nodal-like cells. Over time, an atrial- and ventricular-specific approach has been reported, allowing for the distinction of hiPSCs differentiation [107].

The use of hiPSCs-CMs in studying cardiac ion channel functions surpasses heterologous expression systems, like *Xenopus* oocytes, human embryonic kidney (HEK) cells, and Chinese hamster ovary (CHO) cells. The reproduction of cardiac electrophysiological properties is different in these artificial and animal cell models. HiPSCs-CMs provide a valuable model for studying human heart diseases and serve as a crucial preclinical tool for determining pathomechanisms [1,5,108]. They can identify patient-specific susceptibilities and anticipate some off-target effects, offering an excellent human platform for studying the pathogenic mechanisms of cardiac disorders [109]. As a result, they enable new approaches to precision medicine feasible and are highly effective in the study of channelopathies, such as BrS and CPVT.

Furthermore, hiPSCs-CMs offer the opportunity to use specific cells with the same ethnicity, gender, and genetic background as the donor, thereby allowing for the study patient-specific responses on a cellular level. To ensure meaningful comparisons, it is important that the disease lines match in gender, ethnicity, and age [110].

The introduction of 3D-engineered heart tissues (EHTs) using hiPSCs-CMs has revolutionized the study of channelopathies such as BrS and CPVT. These advancements have addressed the immature characteristics, the well-known issue of immaturity in hiPSCs-CMs field and have led to notable improvements in their functionality and structure. To overcome the challenge of hiPSCs-CMs immaturity, often evident when cardiomyocytes remain in culture for 30 days or less, methods to promote maturation have been employed. These include prolonged culture, the addition of hormones like thyroid hormone, and the use of cellular energy sources such as fatty acids. Co-culture techniques, extracellular matrix, and mechanical or electrical stimulation, along with 3D culture, have also played a pivotal role in achieving a more mature state of the cardiomyocytes [111]. These approaches have enhanced cardiomyocyte maturity and the generation of 3D models that closely resemble native heart tissue. It is important to note that the heart is composed of more than just cardiomyocytes. Other cell types, such as endothelial cells, fibroblasts, pericytes, smooth muscle cells, immune cells, adipocytes, mesothelial cells, and neuronal cells, are also present. Research indicates that cardiomyocytes make up only 25–35% of the heart’s cellular composition, while endothelial cells make up 60% and fibroblasts less than 20% [112]. In specific areas like atrial and ventricular samples, cardiomyocytes form 30–50% of the cells, with endothelial cells and fibroblasts representing 10% and 20%, respectively [113]. Including these diverse cell types in models enhances their physiological accuracy and applicability in studying diseases, as well as in drug testing and cardiotoxicity assessments. Different techniques have been used to create 3D cell cultures, such as the scaffold-free hanging droplet method. In this method, hiPSCs-CMs are placed in droplets on ultra-low attachment plates, optionally with the addition of other cell types like cardiac fibroblasts and endothelial cells [114,115]. This method yields long-term stable 3D models of hiPSC-CMs capable of responding to electrical, pharmacological, and physical stimuli. However, challenges such as partial penetration of Ca^2+^ dyes and retention of fetal-like features, like shortened sarcomeres, have been observed [116]. Another approach involves combining hiPSC-CM with human cardiac fibroblasts and coronary artery endothelial cells, leading to the formation of cardiac spheroids, which more accurately reflect the human heart in vivo [114]. Cardiac microtissues (cMTs) are another scaffold-free product, formed by combining various cell types, including hiPSCs-derived endothelial cells, cardiac fibroblasts, and hiPSC-CMs. These microtissues exhibit mature hiPSC-CM structures, such as elongated tubular myofibrils and T-tubule-like structures, with electrophysiological maturation evidenced by action potential notches. These models have demonstrated their ability to recapitulate disease phenotypes [117]. In scaffold-based 3D cultures, scaffolds composed of decellularized extracellular matrix (ECM) or natural or synthetic polymers are used [118,119,120]. These support the maturation of CMs, with adult heart ECM shown to enhance maturation in both two-dimensional (2D) and 3D cultures [118]. Engineered heart tissue (EHT), where hiPSC-CMs are grown on hydrogel scaffolds, exhibits more mature electrophysiological properties and cardiac ultrastructure, aligning more closely with adult cardiac tissue [121]. Techniques such as progressive stretching of EHT have been employed to further enhance maturation, leading to increased contractility and a more mature excitation/contraction coupling. These EHT models have proven valuable in studying channelopathies, showing reduced arrhythmogenic activity compared to 2D models and mimicking clinical scenarios more accurately [122].

## 7. Human-Induced Pluripotent Stem Cells Derived Cardiomyocytes from Brugada Syndrome Patients

Precision medicine is essential for treating diseases and addressing individual variations in genes and environment. One valuable tool for studying channelopathies is the hiPSCs-CM model. This model has shown promise in producing patient-specific-induced pluripotent stem-cell-derived cardiomyocytes for individuals with BrS (BrS-hiPSCs-CMs). By comparing the electrophysiological properties of the mutant Na_V_1.5 to the wild type, hiPSCs-CMs can provide insights into BrS [123]. Table 1 provides a summary of hiPSCs-CMs models used in BrS-related studies.

In addition, a new model that is not dependent on the patient’s genetic background has been developed. This model uses hiPSC-CMs carrying the CRISPR/Cas9-introduced BrS-variant p.A735V-Na_V_1.5 (g.2204C> T in exon 14 of SCN5A). The electrophysiological properties were studied and compared by a study on mutant A735V-Na_V_1.5 channels heterologously expressed in HEK293T cells. The HEK293T cells were not able to recapitulate the electrophysiological properties, highlighting the need to investigate BrS mechanisms in independent systems [128].

The use of hiPSCs-CMs from a BrS patient with an *SCN5A* variant (p. A226V and p. R1629X) suggested that a potential repolarization deficit may be the underlying mechanism for BrS. The I_Na_- and I_to_-induced action potential changes could be investigated [108]. A study on fever was able to replicate the effects in the human body by subjecting BrS-hiPSCs-CMs carrying a pathogenic variant to heat (c.3148G> A/p. Ala1050Thr) in *SCN5A*. Barajas-Martinez et al. were able to identify the functional impact of the *SCN9A*, *PXDNL*, and *FKBP1B* variants on a polygenic cause of the BrS or early repolarization syndrome (ERS) arrhythmic phenotype. Only carrying all three variants together showed the ERS/BrS phenotype, whereas one or two variants alone did not induce the clinical phenotype [141].

Patch clamp studies with BrS-hiPSCs-CMs without identified variations revealed no difference in sodium current analysis. These findings suggest that ion channel dysfunction, particularly in the cardiac sodium channel, may not be the only sole factor in BrS [134]. Thus far, research has demonstrated that a broad range of variants can be involved and that the precise nature of their influence on channelopathies is still unknown. Precision medicine holds great promise due to the genotype–phenotype associations found in many genes initially linked to BrS or CPVT. One of the short-term goals for hiPSCs-CMs technology is its use for in vitro screening assays. HiPSCs present excellent opportunities for drug development and screening, as they respond differently to cardiotoxic drugs and have the potential to have adverse drug responses. This can be beneficial in the early stages of drug development, as preclinical cardiac drug toxicity is traditionally approached using computational methods or expensive and time-consuming heterologous systems. By utilizing the distinct action potential forms of hiPSCs-CMs, researchers can conduct drug screenings that take into account each subject’s genetic background [142].

Zhu et al. demonstrated that two BrS-hiPSCs-CMs can replicate the characteristics of the BrS single-cell phenotype in response to quinidine [143]. El-Battrawy et al. generated hiPSCs-CMs from a BrS patient with two variants in *SCN1B* (c.629T> C and c.637C> A), which recapitulated some key phenotypic features of BrS, thereby providing a platform for studies on BrS with *SCN1B* variants. The drug efficiency of ajmaline showed a stronger effect on the reduction in amplitude and upstroke velocity in BrS-derived hiPSCs-CMs. Carbachol, on the other hand, increased arrhythmia events and the beating frequency [130]. HiPSCs-CMs derived from a BrS patient with *SCN10A* double variants (c.3803G> A and c.3749G> A) showed that patient-specific hiPSCs-CMs can recapitulate single-cell phenotype features of BrS with *SCN10A* variants. This showed the higher susceptibility of patients to sodium-channel-blocking drugs (ATX II, A-887826, ajmaline) in unmasking BrS [100]. As previously demonstrated, generating BrS-hiPSC-CMs with recurrent ventricular fibrillation while carrying a missense variant in *CACNB2* (c.425C> T/p.S142F) revealed that the variant is pathogenic for this type of BrS and results in a loss of function of L-type calcium channels. The cell model for drug testing reported that quinidine and bisoprolol at low concentrations were useful in reducing arrhythmic events [90].

The mode of action of drugs can be studied using hiPSCs-CMs and functional studies. In order to determine how ajmaline works, a BrS-hiPSCs platform was established. A multielectrode array (MEA) was used to compare cardiac tissue and showed that ajmaline treatment significantly increased the duration of cluster activation–recovery intervals. Patch clamp recordings were conducted to show that ajmaline can block I_Na_ and the rapid delayed rectifier potassium current (I_Kr_) in single CMs isolated from clusters [133].

Implementing an integrated approach that combines both in vivo and in vitro systems, such as hiPSCs-CMs, HL-1 cells, and genetically modified mouse models expressing human-like or mutant genes, could offer a more comprehensive understanding of the electrical phenotype alterations associated with *TBX5* of the Na_V_1.5 variants in BrS. This method would allow for a more detailed exploration of the electrophysiological and molecular differences underlying BrS pathophysiology [144]. Koivumäki et al. have developed a novel in silico model that includes all the essential functional electrophysiology and calcium handling features of hiPSCs-CMs. This virtual cell accurately recapitulates the immature intracellular ion dynamics that are characteristic of hiPSCs-CMs, as measured using in vitro imaging data. The results indicate that hiPSCs-CMs poorly translate the disease-specific phenotypes of BrS, making them less robust and more prone to arrhythmic events compared to adult CMs. While hiPSCs-CMs are functionally more similar to prenatal CMs, they do share some characteristics with adult CMs. This model can serve as a mathematical foundation to improve the translation of human-hiPSCs-CMs [145].

In summary, hiPSC technology has generated interest due to its ability to provide a manageable human-based cellular model. The presence of nearly all endogenous cardiac channels in hiPSCs-CMs cell lines makes studying channelopathies easier.

## 8. Human-Induced Pluripotent Stem-Cells-Derived Cardiomyocytes from Catecholaminergic Polymorphic Ventricular Tachycardia Patients

As mentioned earlier, the use of hiPSC technology has greatly contributed to our understanding of CPVT, a condition characterized by ventricular arrhythmias and irregular handling of calcium ions. This innovative approach has allowed for the creation of patient-specific cardiomyocyte models, revealing the intricate molecular irregularities and ion channel dynamics that are inherent to CPVT. As a result, this has accelerated the development of tailored therapeutic interventions and has advanced the field of precision medicine in the treatment of this condition. Several studies, referenced in Table 2, highlight the usefulness of hiPSCs as in vitro models for CPVT research. The study by Li et al. confirmed that CPVT-hiPSCs maintain a normal karyotype, thereby validating their ability to accurately model the cardiomyocyte pathophysiology associated with CPVT [146]. In the study of Jung et al., hiPSCs-CMs from CPVT patients carrying an *RYR2* S406L variant were generated. HiPSCs-CMs from the patient showed elevated diastolic Ca^2+^ concentrations, reduced SR Ca^2+^ content, and an increased susceptibility to arrhythmias under catecholaminergic stress, compared to control myocytes. These abnormalities were attributed to increased frequency and duration of Ca^2+^ sparks, indicative of aberrant Ca^2+^ handling. Additionally, they found that dantrolene effectively normalized these Ca^2+^ irregularities and mitigated the arrhythmogenic phenotype, which suggests potential therapeutic avenues [147]. Also, a study by Pölönen et al. examined the electrical properties of hiPSCs-CMs from CPVT patients with *RYR2* variants, specifically of the exon 3 deletion (E3D) and L4115F variant. These cells exhibit distinctive action potential characteristics, particularly when exposed to adrenaline, resulting in increased susceptibility to arrhythmias such as DADs and EADs. Notably, E3D-CPVT cells showed the highest susceptibility to arrhythmias. Both carvedilol and flecainide were found to mitigate these arrhythmic effects, though they decreased cell contraction amplitude [148]. This study, again, underscores the value of hiPSCs-CMs for modeling CPVT, offering a profound understanding of the disease’s arrhythmic mechanisms and potential drug testing. To investigate the important role of internal Ca^2+^ reservoirs in the pathogenesis of DAD, Itzhaki et al. generated CPVT-hiPSCs from dermal fibroblasts of a patient with a distinctive M4109R heterozygous point *RYR2* variant. These cells were then differentiated into cardiomyocytes and compared with those derived from healthy controls. Their results indicated significant abnormalities in internal Ca^2+^ handling in CPVT-CMs, contributing to the pathological manifestations in CPVT [104]. In the same context, Preinger et al. investigated the use of hiPSCs-CMs to study CPVT with an emphasis on patient-specific responses to β-blockers. Their research focused on a patient with CPVT who had a unique variant of the *RyR2* gene (Table 2). The patient did not respond well to the β-blocker nadolol but showed significant improvement with flecainide treatment. They generated hiPSCs-CMs from both the patient and controls, which exhibited similar levels of genes involved in excitation-contraction coupling. However, the researchers found that the CPVT patient’s hiPSCs-CMs displayed altered intracellular calcium homeostasis, which validated the disease phenotype. The results showed that while nadolol was ineffective in reducing calcium irregularities, flecainide successfully mitigated these disruptions and aligned with the patient’s response in vivo. This demonstrated that hiPSCs-CMs can replicate important aspects of patient-specific drug responses, highlighting their potential use in developing personalized treatment strategies for CPVT [149]. CPVT is caused by abnormalities in the regulation of intracellular calcium due to variants in either the *RyR2* or *CASQ2* genes. Since CASQ2 and RyR2 have distinct roles in excitation–contraction coupling, Novak et al. hypothesized that there would be differential functional and intracellular calcium discrepancies between cells with variations in these genes. They examined the ultrastructural features and the responses to isoproterenol, caffeine, and ryanodine in both mutated and control hiPSCs-CMs. The results indicated that the mutated CASQ2 and RyR2 cardiomyocytes exhibited less-developed ultrastructural and showed abnormal responses to isoproterenol. These abnormal responses either rendered the treatment ineffective, resulted in arrhythmias, or caused a significant increase in diastolic calcium levels, which contrasted with the positive effects observed in control cells on contraction and relaxation of the heart muscle. Moreover, mutated cardiomyocytes showed altered reactions to caffeine and ryanodine compared to the control group. These results highlight the usefulness of hiPSCs-CMs in understanding the distinct pathophysiological effects of *RyR2* and *CASQ2* variants in CPVT1 and CPVT2 [150]. In a similar vein, Maizels et al. studied hiPSCs derived from a CPVT2 patient (with a D307H-CASQ2 variant) to model the disease and explore its specific mechanisms and pharmacotherapeutic options. Compared to healthy cells, CPVT2-hiPSCs-CMs displayed notable inconsistence in Ca^2+^ transients, increased arrhythmogenicity, and a decreased threshold for Ca^2+^ release events compared to healthy cells. They tested various pharmacological interventions, including β-blockers, flecainide, riluzole, and others, to assess their effectiveness. Carvedilol and JTV-519 showed primary antiarrhythmic actions through SR stabilization. Interestingly, in vitro findings between flecainide and labetalol mirrored clinical observations in the patient. In a recent study, Gao et al. explored CPVT linked to a novel CaM variant using hiPSCs models and biochemical analyses. Derived from a CPVT patient with the *CALM2* p.E46K variant, the hiPSCs-CMs displayed more abnormal electrical and calcium activities compared to control lines, largely attributed to increased Ca^2+^ leakage via the RyR2. This study uncovered that the E46K-CaM variant enhances RyR2 function, especially at lower Ca^2+^ concentration levels, by binding to RyR2 with 10-fold higher affinity than the wild-type CaM, without affecting CaM-Ca^2+^ binding or L-type calcium channel functionality. Notably, the antiarrhythmic drugs nadolol and flecainide effectively mitigated abnormal Ca^2+^ wave activities in these cells, pointing towards potential therapeutic avenues in precision medicine [151]. Ongoing research using hiPSCs-CMs holds great promise in developing personalized treatments and expanding our understanding of CPVT. In conclusion, hiPSCs-CMs have proven to be an effective model for studying the CPVT2 phenotype, providing valuable insights into the disease’s mechanisms and potential patient-specific treatments [152].

## 9. Functional Studies in hiPSC-CMs

For molecular and functional analysis, single cardiomyocytes can be isolated after 40 days of maturing in culture as clusters of beating cells. Several techniques can be used for functional studies, e.g., patch clamp, immunostaining, MEA, and optical mapping (Figure 3). The patch clamp technique is commonly used to analyze channelopathies, providing information about generated action potential and the ion currents. The hiPSCs-CMs can be distinguished through their action potential morphology and expression of specific atrial or ventricular proteins. HiPSCs-CMs application possibilities are very promising. They act as a great human platform to understand the pathogenic mechanisms of cardiac disorders through functional analysis. An overview of the functional studies used for hiPSCs-CMs analysis is described below.

## 10. Patch Clamp

The patch clamp method is a traditional approach that provides intracellular electrical recordings from cardiomyocytes. The electrodes can record the membrane potential by penetrating the cell membrane, thus providing detailed data on ion currents [154]. The perforated patch clamp method is a variation of the patch clamp method, used to assess the total activity of ion channels. The combination of automated patch clamp and the perforated patch clamp approach with hiPSCs-CMs provides excellent precision [155]. It allows the recording of multiple ion current components and APs in many cells at the same time under prolonged conditions.

## 11. Mechanics

Atomic force microscopy (AFM) is a very-high-resolution type of scanning probe microscopy. This enables AFM to be applied as a micromechanical transducer or mechanical nanosensor for biosensor construction. The mechanical forces can be measured by AFM and combined as a putative read out system for hiPSC-CMs. Uniformly sized embryonic bodies containing hiPSC-CMs were integrated into the AFM force-sensing platform by Pesl et al. [156] This enables a high-fidelity contraction pattern in complex conditions and cellular models when combined with the beat rate frequency. It offers a chance to characterize cardiac cell clusters, evaluate the effects of model medications, and separate electrophysiological data from mechanical triggers [157].

To quantify the contraction function in single hiPSC-CMs at high throughput and over time, contractile function tracking is required. CONTRAX offers a quantitative monitoring of the contractile dynamics over time. The 3D TFM has the advantage of being more comprehensive and precise. However, because of the plane-by-plane imaging modality currently in use, 3D imaging has limited time resolution, which makes it challenging to apply to fast-contracting cells like cardiomyocytes and to achieve higher throughput. Another difficult part of the computing requirement is the sheer amount of data to process in such circumstances. Applying the workflow in long-term longitudinal studies raises additional potential concerns about the stability of the mechanical properties of the hydrogel substrates and the ECM micropatterns, and appropriate controls must be used. Pardon et al. proved how effective CONTRAX can be at tracking changes in tractile phenotypes over time [158].

## 12. Multielectrode Array

Multielectrode arrays (MEAs) bring a high-throughput, noninvasive, and nonterminal method for the investigation of electrophysiological properties of cardiomyocytes. It is based on a plated dish containing electrodes that measure the field potential [159,160]. Since the intracellular action potential and extracellular field potential are correlated, the MEA system records both stimulated and spontaneous electrical activity [161]. It is used for high-throughput drug screening in hiPSCs-CMs [162,163].

## 13. Optical Mapping

Optical mapping is a set of techniques used to observe electrophysiological activities in cardiomyocytes using light-based methods.

In hiPSC-CMs, optical mapping can be employed to observe the reentry mechanisms in the tissue [164]. Nontransparent zebrafish ECG recordings have been used to study BrS as well [165]. Optical mapping allows for the examination of various electrophysiological parameters, enabling the measurement of repolarization changes in channelopathies in the presence of drugs. It can be used as an alternative to, or in conjunction with, patch clamp techniques [166].

## 14. Impedance

Impedance assays are a noninvasive and label-free method with a high-throughput rate. Cells are seeded on a plate that contains electrodes on the bottom. Applying a weak current leads to attachment of the cells on the electrodes. This method is based on the indirect detection of the hiPSCs-CMs beating. The mechanical movements are detected by the shift in the impedance signal after they have passed through the electrodes [167].

## 15. Limitations

Though hiPSCs technology was a milestone in disease modeling, it is still a limited approach. One of the well-known disadvantages of hiPSCs-CMs is the immaturity and the heterogeneity of the cells [145,168,169]. HiPSCs-CMs have altered gene expression, reduced sarcomere organization, aberrant morphology, and fewer T-tubules due to their immaturity. Compared to adult CMs, hiPSC-CMs exhibit less robustness and a higher likelihood of arrhythmic events due to their poor translation of the disease-specific phenotypes of BrS and CPVT.

One of the main drawbacks of hiPSC-CMs is their immaturity; in single-cell studies, numerous elements such as cellular interaction, nerve, and hormone regulations are not taken into account [170]. The limitations of hiPSC-CMs, which exhibit a relatively immature fetal-like phenotype, can be overcome by extending the cell culture time, using a medium containing galactose and fatty acids, electrical field stimulation, electric pacing and mechanical stimulation, ECM, and 3D cardiac tissue with electric stimulation. Regardless of the previous successes and possibilities, they are still limitations of hiPSCs-CMs in terms of their clinical approaches. Reprogramming hiPSCs can result in changes in epigenetic modifications [171]. Although the genetic background can be reconstituted, years of environmental influences of different lifestyles, unfortunately, cannot be.

It is worth noting that ion channel expression, cytoskeleton proteins, and morphology of the hiPSCs-CMs all indicate signs of immaturity. Another disadvantage is the lack of ion channel expression and cytoskeleton proteins, as well as the missing complexity of cardiac tissue as a 2D model [172]. To address this issue, models such as EHTs, cell sheets or microtissues have been developed to capture the complexity of the tissue [117,173,174]. The advancement of organ on-a-chip platforms will revolutionize cardiac modeling and improve the predictability of this technology [175].

In this review, we underscored the significance of hiPSC-CMs for investigating patient-specific molecular and genetic mechanisms underlying cardiac channelopathies. Nonetheless, we also acknowledge the advantages of alternative models, such as engineered heart tissues (EHTs) and human living myocardial slices (hLMSs). EHTs, characterized by their advanced maturation in three-dimensional culture systems, more accurately replicate the behavior of adult cardiomyocytes, rendering them particularly suitable for long-term drug screening and functional assessments [176]. In contrast, while hLMSs preserve the native cardiac architecture and provide substantial physiological relevance, their limited viability and lack of patient specificity present notable challenges [177]. To address these limitations, integrating hiPSC-CMs with EHTs or hLMSs could significantly enhance the understanding of cardiac disease mechanisms, offering a more comprehensive research approach.

The potential of hiPSCs-CMs as experimental tools in drug development and cardiovascular research has been demonstrated. However, their usefulness as a human platform is still limited due to their differences from adult human cardiomyocytes. Thus far, it is not yet possible to study diseases through the involvement of complex genetic interactions and environmental factors. Identifying genetic modifiers has been challenging due to their unpredictable nature. However, genome editing methods like CRISPR/Cas9 genome editing, zinc finger nucleases, and transcription activator-like effector nuclease can also encounter issues.

Thus far, research and development have demonstrated that by improving the concept, some of the limitations can be overcome.

## 16. Conclusions

In conclusion, despite some drawbacks, hiPSCs-CMs represent an excellent model for the exploration of inherited cardiac channelopathies. Channelopathies are a rather unexplored group of diseases, which together account for a large proportion of cardiac diseases. HiPSCs technology allows us to create patient-specific approaches that are crucial for these diseases with different genotype–phenotype correlations. They can serve as human platforms and are quicker and more efficient than animal models, for example. As previous innovations and advancements in hiPSC technology have shown, more research is needed to ensure that hiPSC cell lines can be used as cohort models in the future to study channelopathies.

## Figures and Tables

**Figure 1 ijms-25-12034-f001:**
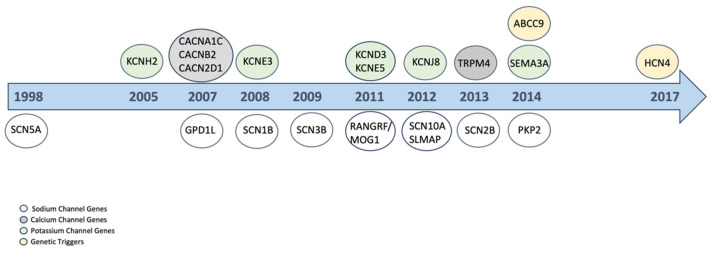
Genes involved in Brugada syndrome. Chronology of the discovery of the genes linked to Brugada syndrome. The genes are classified as sodium channel, calcium channel, potassium channel genes, and genetic triggers based on their molecular interactions regarding Brugada syndrome.

**Figure 2 ijms-25-12034-f002:**
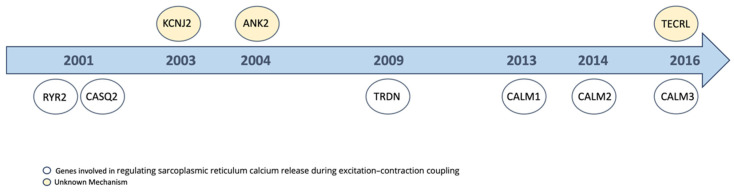
Genes involved in catecholaminergic polymorphic ventricular tachycardia. Chronology of the discovery of the genes linked to catecholaminergic polymorphic ventricular tachycardia. The genes are classified as genes involved that regulate sarcoplasmic reticulum calcium release during excitation–contraction coupling and genes with an unknown mechanism leading to catecholaminergic polymorphic ventricular tachycardia.

**Figure 3 ijms-25-12034-f003:**
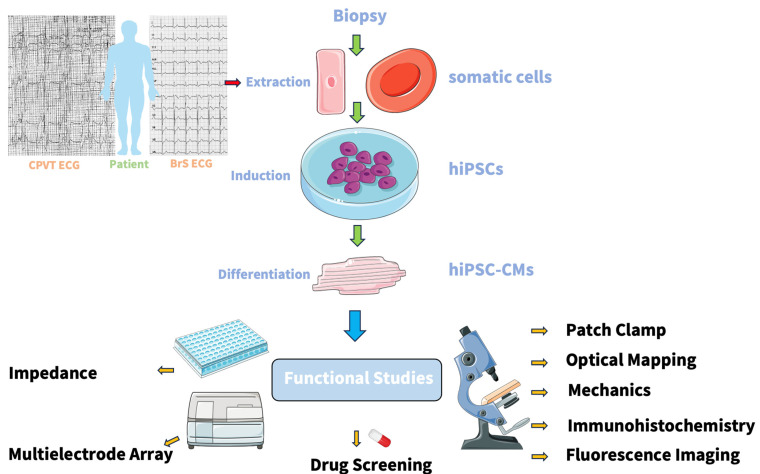
The use of several techniques to carry out functional studies on hiPSCs-CMs. Patients are typed to the respective channelopathies based on their electrocardiogram (ECG). Biopsies are taken from patients suffering Brugada syndrome (BrS) or catecholaminergic polymorphic ventricular tachycardia (CPVT). Somatic cells are extracted from patients’ skin or blood biopsies. The somatic cells are reprogrammed to human-induced pluripotent stem cells (hiPSCs). The differentiation into hiPSCs-cardiomyocytes (hiPSCs-CMs) is induced by culturing under specific conditions. hiPSCs-CMs can be used for modeling channelopathies. Functional studies on hiPSCs-CMs include patch clamp, optical mapping, mechanics, immunohistochemistry, fluorescence imaging drug screening, multielectrode array, and impedance assays. Image created using Smart Servier Medical Art.

**Table 1 ijms-25-12034-t001:** Summary of hiPSC-CMs models derived from BrS patients.

Reference	Gene	OMIM	Variant	Phenotype
Davis et al., 2012 [124]	*SCN5A*	601144/ 603830	c.5537insTGA (p.1795insD)	Reduced I_Na_; reduced V_max._
Kosmidis et al., 2016 [125]	*SCN5A*	600163	c.4912C> T (p.R1638X) c.468G> A (p.W156X)	Reduced I_Na_; reduced V_max._
Liang et al., 2016 [126]	*SCN5A*	600163.006	c.2053G> A (p.R620H) and c.2626G> A (p.R811H) c.4190delA (p.K1397Gfs*5)	Reduced I_Na_; reduced V_max_; abnormal Ca^2+^ transients.
Ma et al., 2018 [108]	*SCN5A*	600163.004	c.677> T (p.A226V) and c.4885C> T (p.R1629X) c.4859C> T p.T1620M	Reduced I_Na_; reduced V_max._
Selga et al., 2018 [127]	*SCN5A*	601144	c.1100G> A (p.367H)	Reduced I_Na_ density.
De la Roche et al., 2019 [128]	*SCN5A*	601144	c.2204C> T (p.A735V)	Reduced I_Na_; reduced V_max_; rightward shift of the steady-state activation curve.
Okata et al., 2016 [129]	*SCN5A*	600163	c.5349G> A (p.E1784K)	Reduced I_Na._
El-Battrawy, Albers et al., 2019 [100]	*SCN10A*	604427	c.3749G> A (p.R1250Q) and c.3808G> A (p.R1268Q)	Reduced I_Na_ density; reduced V_max._
El-Battrawy, Müller et al., 2019 [130]	*SCN1B*	600235	c.629T> C (p.L210P) and c.637C> A (p.P213T)	Reduced I_Na_ density; reduced V_max_; rightward shift of the steady-state activation curve.
El-Battrawy et al., 2021 [131]	*CACNB2*	600003	c.1439C> T/p.S480L	Reduced APD; reduced I_CaL._
Belbachir et al., 2019 [132]	*RRAD*	179503	p.R211H	Reduced I_Na_ density; reduced V_max_; prolonged AP; increased EAD; reduced I_CaL_.
Cerrone et al., 2014 [22]	*PKP2*	602861	c.1904G> A (R635Q)	Reduced I_Na._
Miller et al., 2017 [133]	Undefined		Undefined	Reduced I_Na_ density.
	Undefined		Undefined	
	*PKP2*	602861	c.302G> A (p.R101H)	
Veerman et al., 2016 [134]	Undefined		Undefined	Reduced I_Na_; reduced V_max._
	Undefined		Undefined	
	*CACNA1C*	114205	Int19 position -7 (benign)	
Li et al., 2020 [135]	*SCN5A*	600163	p.S1812X11	Reduced I_Na_ density; reduced V_max_; increased I_CaL._
Simons et al., 2022 [136]	*SCN5A*	600163	c.4813+3_4813+6dupGGGT	-
Kashiwa et al., 2023 [137]	*CACNA1C*	114205.0019	E1115K	Longer APD; increased EAD; reduced I_CaL._
Li et al., 2023 [138]	*SCN5A*	600163	c.3148G> A/p.Ala1050Thr	Reduced I_Na_; reduced V_max._
Zhong et al., 2022 [90]	*CACNB2*	600003	c.425C> T/p.S142F	Reduced I_CaL._
Cai et al., 2023 [139]	*SCN5A*	600163	T1788fs	Reduced I_Na_; reduced V_max._
Kamga et al., 2021 [140]	*SCN5A*	600163	Undefined	Reduced I_Na_; reduced V_max._

**Table 2 ijms-25-12034-t002:** Summary of hiPSC-CMs models derived from CPVT patients.

Reference	Gene	OMIM	Variant	Phenotype
Jung et al., 2012 [147]	*RYR2*	180902.0007	p.(Ser406Leu) p.(Pro2328Ser)	Increased diastolic [Ca^2+^]; reduced SR Ca^2+^; increased DAD.
Itzhaki et al., 2012 [104]	*RYR2*	180902	p.(Met4109Arg)	Increased DAD; store-overload-induced Ca^2+^ release.
Preiniger et al., 2016 [149]	*RYR2*	180902	p.(Leu3741Pro)	Increased diastolic [Ca^2+^]; reduced SR Ca^2+^.
Novak et al., 2015 [150]	*RYR2*	180902	p.(Ile4587Val)	Increased diastolic [Ca^2+^]; reduced SR Ca^2+^.
Novak et al., 2015 & Nijak et al., 2021 [150,153]	*CASQ2*	114251	p.(Asp307His)	Increased diastolic [Ca^2+^]; reduced SR Ca^2+^; increased DAD.
Pölönen et al., 2020 [148]	*RYR2*	180902.0011	Exon 3 Deletion (E3D) L4115F	Reduced APD; increased V_max_; increased diastolic [Ca^2+^].
Maizels et al., 2017 [152]	*CASQ2*	114251	p.(Asp307His)	Increased EAD; reduced threshold for store-overload-induced Ca^2+^ release.
Gao et al., 2023 [151]	*CALM2*	114182	p.E46K	Reduced SR Ca^2+^.
Kamga et al., 2021 [140]	*RYR2*	180902	Undefined	Increased diastolic [Ca^2+^]; reduced SR Ca^2+^.
Li et al., 2021 [146]	*RYR2* *CASQ2*	180902 114251.0001	p.A2254V D307H	-

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
