# Peer review of "The Role of Human-Induced Pluripotent Stem Cells in Studying Cardiac Channelopathies"

_ijms, 2024, doi:10.3390/ijms252212034_

Round 1
Reviewer 1 Report
Comments and Suggestions for Authors
In the review article 'The role of human-induced pluripotent stem cells in studying cardiac channelopathies' submitted by Begovic et al. to IJMS, the authors summarize the use of iPSCs in the context of cardiac channelopathies. The topic of this review article is interesting, but in my view some major points should be addressed before it should be accepted for publication in IJMS.
Minor Points:
1.) Not all abbreviations like CRISPR/Cas9 have been explained.
2.) Some typos have to be corrected (e. g. Page 2, Line 1, ‘Inherited’ instead of ‘inherited’.
3.) Please add OMIM identifiers for each gene and each genetic disease, which you mention.
4.) At the end of your introduction, I would shortly mention and explain that also different CM cause SCD. Similarly, in the context of cardiomyopathies, several iPSC-derived cardiomyocytes studies revealed important insights into SCD. The review article ‘Human Induced Pluripotent Stem-Cell-Derived Cardiomyocytes as Models for Genetic Cardiomyopathies’ would be a good starting point for this topic.
Major Points:
1.) Page 2: I would shortly explain that PKP2 is the major ARVC/ACM gene. See ‘Mutations in the desmosomal protein plakophilin-2 are common in arrhythmogenic right ventricular cardiomyopathy ‘ (Nature Genetics 2004). Interestingly, there is an overlap between ARVC/ACM and Brugada syndrome. In my view, you should discuss here also shortly this overlap.
2.) Please prepare a summarizing figure about the genes involved in Brugada syndrome and also in CPVT and please explain their cellular and molecular function in more detail.
3.) A time line of the discovered genes involved in chanellopathies would be also helpful for the reader.
4.) I am wondering why the authors have not included Long QT and Short QT in this review article. I would not ignore this.
5.) Paragraph 4: Human induced pluripotent stem cells (hiPSCs): What is with transfection of episomal plasmids and what is with transfection of mRNA encoding the Yamanaka factors. I would not ignore these techniques, since both have less problems in the context of biosafety issues. Please explain also these biosafety issues here.
6.) In the discussion, I am wondering why the authors have not compared iPSC-derived cardiomyocytes with other techniques like EHTs (engineered heart tissue) and human living myocardial slices? I would address this point.
7.) I suggest also to include contraction measurements using atomic force microscopy and the edge-detection (contraction) system as a putative read out for iSPC-derived cardiomyocytes. Please include this in your review article.
Since some important points are not reviewed in this article, I suggest a major revision. However, the topic of this review article is interesting and highly relevant within the research community.
Author Response
Reviewer 1
In the review article 'The role of human-induced pluripotent stem cells in studying cardiac channelopathies' submitted by Begovic et al. to IJMS, the authors summarize the use of iPSCs in the context of cardiac channelopathies. The topic of this review article is interesting, but in my view some major points should be addressed before it should be accepted for publication in IJMS.
Response: We appreciate the reviewer's comments and will address each point systematically.
Minor Points:
1.) Not all abbreviations like CRISPR/Cas9 have been explained.
Response: We have introduced all abbreviations now.
2.) Some typos have to be corrected (e. g. Page 2, Line 1, ‘Inherited’ instead of ‘inherited’.
Response: Thank you very much for your note. We have corrected all typos throughout the article.
3.) Please add OMIM identifiers for each gene and each genetic disease, which you mention.
Response: We greatly appreciate your suggestions for enhancements and have added OMIM identifiers for each gene as recommended.
4.) At the end of your introduction, I would shortly mention and explain that also different CM cause SCD. Similarly, in the context of cardiomyopathies, several iPSC-derived cardiomyocytes studies revealed important insights into SCD. The review article ‘Human Induced Pluripotent Stem-Cell-Derived Cardiomyocytes as Models for Genetic Cardiomyopathies’ would be a good starting point for this topic.
Response: We appreciate the insightful suggestion and have incorporated a concise discussion at the conclusion of the introduction regarding the various cardiomyopathies that can precipitate sudden cardiac death (SCD). Distinct types of cardiomyopathies, encompassing both genetic and acquired variations, significantly contribute to the risk of SCD. Moreover, we recognize the critical role of induced pluripotent stem cell (iPSC)-derived cardiomyocytes in enhancing our understanding of the mechanisms underlying SCD. The review article "Human Induced Pluripotent Stem-Cell-Derived Cardiomyocytes as Models for Genetic Cardiomyopathies" has been cited to provide a foundational perspective on this topic, highlighting how these models can elucidate the underlying pathophysiology of SCD within the context of cardiomyopathies.
Major Points:
1.) Page 2: I would shortly explain that PKP2 is the major ARVC/ACM gene. See ‘Mutations in the desmosomal protein plakophilin-2 are common in arrhythmogenic right ventricular cardiomyopathy ‘ (Nature Genetics 2004). Interestingly, there is an overlap between ARVC/ACM and Brugada syndrome. In my view, you should discuss here also shortly this overlap.
Response: We appreciate the valuable guidance and have expanded our discussion on plakophilin-2 (PKP2), a significant gene associated with arrhythmogenic right ventricular cardiomyopathy (ARVC/ACM).
“Plakophilin-2 (PKP2) plays a crucial role in cell-cell adhesion and is frequently mutated in inherited cardiac diseases, including arrhythmogenic right ventricular cardiomyopathy (ARVC/ACM) and BrS.22,31 Notably, a phenotypic overlap exists between ARVC/ACM and BrS, suggesting that these conditions may represent different manifestations of a common underlying pathological process.32 Cerrone et al. first reported cases of BrS wihtout overt structural cardiomyopathy in patients carrying PKP2 variants, indicating a potential link between these conditions.22 While recent evidence-based studies have identified SCN5A as the primary gene associated with BrS, the role of PKP2 should not be overlooked.33,34 Several preclinical studies in ARVC showed a significant decrease of the peak sodium channel current consistent with the finding in the BrS. The pleiotropic effects of PKP2 mutations provide crucial insights into how genetic alterations in different proteins interact at the connexome level, influencing both action potential dynamics and the structural integrity of the myocardium.”
2.) Please prepare a summarizing figure about the genes involved in Brugada syndrome and also in CPVT and please explain their cellular and molecular function in more detail.
3.) A timeline of the discovered genes involved in chanelopathies would be also helpful for the reader.
Response: We appreciate your insightful suggestions. We acknowledge the significance of summarizing the genes implicated in Brugada syndrome (BrS) and catecholaminergic polymorphic ventricular tachycardia (CPVT), along with their respective cellular and molecular functions. In response to this consideration, we have created two new figures that present a comprehensive timeline of gene discoveries related to channelopathies. Each figure systematically categorizes genes according to their molecular mechanisms of action, thereby facilitating a deeper understanding of their roles in these conditions.
Figure 1. Genes involved in Brugada Syndrome.
Figure 1. Genes involved in Brugada Syndrome. Chronology of the discovery of the genes linked to Brugada syndrome. The genes are classified as sodium channel, calcium channel, potassium channels genes and genetic triggers, based on their molecular interactions regarding the Brugada syndrome.
Figure 2. Genes involved in Catecholaminergic Polymorphic Ventricular Tachycardia.
Figure 2. Genes involved in Catecholaminergic Polymorphic Ventricular Tachycardia. Chronology of the discovery of the genes linked to catecholaminergic polymorphic ventricular tachycardia. The genes are classified as genes that regulate sarcoplasmic reticulum calcium release during excitation-contraction coupling and genes with an unknown mechanism leading to catecholaminergic polymorphic ventricular tachycardia.
4.) I am wondering why the authors have not included Long QT and Short QT in this review article. I would not ignore this.
Response: Thank you for your valuable feedback. We concur that Long QT syndrome (LQTS) and Short QT syndrome (SQTS) are significant cardiac channelopathies that warrant attention in this review. In response, our article has expanded to include a concise discussion of both LQTS and SQTS, detailing their clinical significance, genetic foundations, and the associated risk of sudden cardiac death (SCD).
We have added this new paragraph to our review article:
“LQTS and SQTS syndrome are significant genetic cardiac channelopathies that markedly increase the risk of arrhythmias and sudden cardiac death (SCD). LQTS is characterized by delayed ventricular repolarization, resulting in a prolonged QT interval, which predisposes affected individuals to life-threatening arrhythmias such as torsades de pointes. In contrast, SQTS is distinguished by accelerated repolarization, leading to a shortened QT interval and heightened susceptibility to both atrial and ventricular arrhythmias. Both conditions arise from gene mutations that encode ion channels, ultimately disrupting cardiac electrophysiology and rhythm stability.
Recent studies using hiPSC-derived cardiomyocytes have elucidated the mechanisms that are underlying LQTS and SQTS. These stem cell models provide a valuable platform for investigating patient-specific gene-mutations and the effects of pharmacological interventions, thereby enhancing our understanding of and treatment of these syndromes.1–3
In the cardiovascular system, ion channels play a crucial role in various aspects of cardiac function, including rhythmicity and contractility. When there is an alteration in these ion channels, it increases the risk of atrial and ventricular arrhythmic events, which can predispose individuals to sudden cardiac death (SCD). Inherited arrhythmia syndromes, which encompass Inherited cardiac channelopathies, account for more than 30% of SCD cases in young individuals without any underlying structural heart disease at a young age.4 Distinct types of cardiomyopathies, encompassing both genetic and acquired variations, significantly contribute to the risk of SCD. Moreover, we recognize the critical role of iPSC-derived cardiomyocytes in enhancing our understanding of the mechanisms underlying SCD. The review article "Human Induced Pluripotent Stem-Cell-Derived Cardiomyocytes as Models for Genetic Cardiomyopathies" has been cited to provide a foundational perspective on this topic, highlighting how these models can elucidate the underlying pathophysiology of SCD within the context of cardiomyopathies.5–10
- Long QT Syndrome and Short QT Syndrome
LQTS and SQTS are congenital cardiac channelopathies that disrupt the heart's electrical repolarization phase, resulting in abnormal cardiac rhythms and an increased risk of SCD. In LQTS, delayed ventricular repolarization prolongs the QT interval, thereby elevating the risk of torsades de pointes, a potentially fatal ventricular arrhythmia. This condition is frequently attributed to mutations in genes such as KCNQ1, KCNH2, and SCN5A, which encode potassium and sodium channels.
In contrast, SQTS is characterized by an abnormally shortened QT interval due to accelerated ventricular repolarization, which is caused by gain-of-function mutations in potassium channels (KCNH2, KCNQ1 and KCNJ2) or loss-of-function mutations in calcium channels (CACNA1C/CACNB2) or affecting chloride bicarobate transporters encoded by SLC4A3. Such mutations predispose patients to atrial fibrillation, ventricular arrhythmias, and SCD. While LQTS results in delayed repolarization, SQTS facilitates its acceleration; both conditions can lead to significant arrhythmogenic disturbances.
A comprehensive understanding of these syndromes highlights the critical role of ion channels in maintaining cardiac rhythm. Furthermore, it emphasizes the challenges associated with diagnosing and managing these life-threatening conditions, which often necessitate individualized therapeutic strategies such as b-blockers, implantable cardioverter-defibrillator (ICD) implantation, or anti-arrhythmic medications. Therefore, LQTS and SQTS are integral to advancing our understanding of cardiac electrophysiology and refining strategies for the prevention of SCD.”
5.) Paragraph 4: Human induced pluripotent stem cells (hiPSCs): What is with transfection of episomal plasmids and what is with transfection of mRNA encoding the Yamanaka factors. I would not ignore these techniques, since both have less problems in the context of biosafety issues. Please explain also these biosafety issues here.
Response: We appreciate the reviewer’s insightful comments and have expanded our discussion to incorporate additional reprogramming techniques for generating human induced pluripotent stem cells (hiPSCs).
“Episomal plasmids present a non-integrating alternative to viral methods, thereby minimizing biosafety concerns. These plasmids facilitate the expression of reprogramming factors (OCT4, SOX2, NANOG, etc.) without the risk of genomic integration. Their episomal nature permits removal from the cells, thereby enhancing safety for clinical applications. 90,1,100-101”
and
“Furthermore, mRNA transfection provides a vector-free approach that eliminates risks associated with residual vector traces. This technique also offers improved reprogramming speed and genomic integrity, rendering it suitable for the generation of clinical-grade hiPSCs. While blood-derived iPSCs show promise, their clinical application necessitates overcoming challenges related to mass production and reprogramming efficiency. By employing these non-integrating techniques, it is possible to address biosafety concerns and enhance the clinical potential of hiPSC-derived cells.102 “
6.) In the discussion, I am wondering why the authors have not compared iPSC-derived cardiomyocytes with other techniques like EHTs (engineered heart tissue) and human living myocardial slices? I would address this point.
Response: We sincerely appreciate your significant notification. We must address the comparison of EHTs and hiPSC-CMs in the discussion.
In this review, we underscore the significance of hiPSC-CMs for investigating patient-specific molecular and genetic mechanisms underlying cardiac channelopathies. Nonetheless, we also acknowledge the advantages of alternative models, such as engineered heart tissues (EHTs) and human living myocardial slices (LMS). EHTs, characterized by their advanced maturation in three-dimensional culture systems, more accurately replicate the behavior of adult cardiomyocytes, rendering them particularly suitable for long-term drug screening and functional assessments.185 In contrast, while hLMS preserve the native cardiac architecture and provide substantial physiological relevance, their limited viability and lack of patient specificity present notable challenges.186 To address these limitations, the integration of hiPSC-CMs with EHTs or hLMS could significantly enhance the understanding of cardiac disease mechanisms, thereby offering a more comprehensive research approach.
7.) I suggest also to include contraction measurements using atomic force microscopy and the edge-detection (contraction) system as a putative read out for iSPC-derived cardiomyocytes. Please include this in your review article.
Response: Thank you very much for your guidance. We added details regarding the AFM and edge-detection.
“Atomic force microscopy (AFM) is a very high-resolution type of scanning probe microscopy. This enables AFM to be applied as a micromechanical transducer or mechanical nanosensor for biosensor construction. The mechanical forces can be measured by AFM and combined as a putative read out system for hiPSC-CMs. Uniformly sized embryonic bodies containing hiPSC-CMs were integrated into the AFM force-sensing platform by Pesl et al. 145. This enables a high-fidelity contraction pattern in complex conditions and cellular models when combined with the beat rate frequency. It offers a chance to characterize cardiac cell clusters, evaluate the effects of model medications, and separate electrophysiological data from mechanical triggers.160”
To quantify the contraction function in single hiPSC-CMs at high throughput and over time contractile function tracking is required. CONTRAX offers a quantitative monitoring of the contractile dynamics over time. The 3D TFM has the advantage of being more comprehensive and precise. However, because of the plane-by-plane imaging modality currently in use, 3D imaging has limited time resolution, which makes it challenging to apply to fast-contracting cells like cardiomyocytes and to achieve higher throughput. Another difficult part of the computing requirement is the sheer amount of data to process in such circumstances. Applying the workflow in long-term longitudinal studies raises additional potential concerns about the stability of the mechanical properties of the hydrogel substrates and the ECM micropatterns and appropriate controls must be used. Pardon et al. has proven how effective CONTRAX can be at tracking changes in tractile phenotypes over time.161 “

Reviewer 2 Report
Comments and Suggestions for Authors
HiPSCs-CMs as a model had been used to study the pathophysiological mechanisms of various channelopathies and drug screening for many years, but still there are many limitations. The topic of this review manuscript which highlights the role of hiPSCs-CMs in understanding the pathomechanism of Brugada syndrome (Brs) and catecholaminergic polymorphic ventricular tachycardia (CPVT) is interesting. However, the manuscript has too many peripheral subjects, for example, the too detailed information of clinical backgrounds of Brs and CPVT, the methods of patch clamp and optical mapping, et al. The authors should focus more on the advantages and disadvantages of HiPSCs-CMs as models to study the pathomechanisms of Brs and CPVT and how these models can be utilized for drug screening.
1. In the introduction, the other subtypes of channelopathies and the reasons for the authors to highlight Brs and CPVT in this review should be introduced.
2. Too tedious clinical backgrounds of Brs and CPVT.
3. In Tables 1 and 2, it is better to have the functional phenotypes of the different mutations, so the readers can better understand whether the hiPSCs-CMs be a good disease model.
4. The methods to generate Hipsc line and the detailed information of the patch clamp and the optical mapping are less relevant with the topic.
5. The most paragraphs are too long for readers to follow.
6. In page 2, the statement “However, in a bevy of cases of patients carrying variants in the SCN5A or other genes, the causal relationship has not been proved yet” needs reference.
Author Response
Reviewer 2
Comments and Suggestions for Authors
HiPSCs-CMs as a model had been used to study the pathophysiological mechanisms of various channelopathies and drug screening for many years, but still there are many limitations. The topic of this review manuscript which highlights the role of hiPSCs-CMs in understanding the pathomechanism of Brugada syndrome (Brs) and catecholaminergic polymorphic ventricular tachycardia (CPVT) is interesting. However, the manuscript has too many peripheral subjects, for example, the too detailed information of clinical backgrounds of Brs and CPVT, the methods of patch clamp and optical mapping, et al. The authors should focus more on the advantages and disadvantages of HiPSCs-CMs as models to study the pathomechanisms of Brs and CPVT and how these models can be utilized for drug screening.
Response: We thank the reviewer for the evaluation. We shortened the clinical background of BrS and CPVT.
- In the introduction, the other subtypes of channelopathies and the reasons for the authors to highlight Brs and CPVT in this review should be introduced.
Response: Thank you for your valuable comment. We have revised the introduction accordingly:
"The most common cardiac channelopathies include Long QT syndrome (LQTS) and Short QT syndrome (SQTS), in addition to conditions like Early repolarization syndrome, Catecholaminergic polymorphic ventricular tachycardia (CPVT), Brugada syndrome (BrS), and isolated progressive cardiac conduction disease. In this review, we focus on BrS and CPVT due to their rarity and the significant insight they provide into the genetic and molecular mechanisms underlying less understood channelopathies."
- Too tedious clinical backgrounds of Brs and CPVT.
Response: Thank you very much for your notice. We shortened the clinical background of BrS and CPVT.
- In Tables 1 and 2, it is better to have the functional phenotypes of the different mutations, so the readers can better understand whether the hiPSCs-CMs be a good disease model.
Response: We sincerely appreciate your advice. The tables are changed.
- The methods to generate Hipsc line and the detailed information of the patch clamp and the optical mapping are less relevant with the topic.
Response: We welcome your recommendation and adjust the manuscript.
- The most paragraphs are too long for readers to follow.
Response: Thank you very much for your impression. We comprehensed the clinical backgrounds and the methods.
- In page 2, the statement “However, in a bevy of cases of patients carrying variants in the SCN5A or other genes, the causal relationship has not been proved yet” needs reference.
Response: We greatly appreciate your suggestion and will prove our sentence with references.
“However, in a bevy of cases of patients carrying variants in the SCN5A or other genes, the causal relationship has not been proved yet.20,21 “
(15) Smits, J. P. P.; Eckardt, L.; Probst, V.; Bezzina, C. R.; Schott, J. J.; Remme, C. A.; Haverkamp, W.; Breithardt, G.; Escande, D.; Schulze, -Bahr Eric; LeMarec, H.; Wilde, A. A. M. Genotype-Phenotype Relationship in Brugada Syndrome: Electrocardiographic Features Differentiate SCN5A-Related Patients from Non–SCN5A-Related Patients. Journal of the American College of Cardiology 2002, 40 (2), 350–356. https://doi.org/10.1016/S0735-1097(02)01962-9.
(16) Probst, V.; Wilde, A. A. M.; Barc, J.; Sacher, F.; Babuty, D.; Mabo, P.; Mansourati, J.; Le Scouarnec, S.; Kyndt, F.; Le Caignec, C.; Guicheney, P.; Gouas, L.; Albuisson, J.; Meregalli, P. G.; Le Marec, H.; Tan, H. L.; Schott, J.-J. SCN5A Mutations and the Role of Genetic Background in the Pathophysiology of Brugada Syndrome. Circulation: Cardiovascular Genetics 2009, 2 (6), 552–557. https://doi.org/10.1161/CIRCGENETICS.109.853374.

Round 2
Reviewer 1 Report
Comments and Suggestions for Authors
Congratulation! The revised version of this nice review article is really convincing. The authors have improved their manuskript significantly. Therefore, I suggest to accept this review article for publication without any restrictions. Congratulation!
Author Response
We appreciate your comments.
Reviewer 2 Report
Comments and Suggestions for Authors
In the tables 1 and 2, the phenotype should be the summarized electrophysiological functions of the relevant channels, instead of "Brs" or "CPVT". The ion channel function changes are not equal to the disease.
Author Response
We appreciate your comments. We added the ion channel changes in table 1 and table 2 as required.
